# Host–virus association databases as tools for understanding viral spillover at varying scales

Imogen C. Lindsley [1�] *, Maya M. Juman[1उ], Stephanie N. Seifert[2], Rory Gibb[3], Gregory F. Albery[4], Freya Jephcott[5], Olivier Restif[1]

1 Department of Veterinary Medicine, University of Cambridge, Cambridge, United Kingdom, 2 Paul G. Allen School for Global Health, Allen Center, Washington State University, Pullman, Washington, United States of America, 3 Department of Genetics, Evolution & Environment, University College London, London, United Kingdom, 4 Department of Zoology, Trinity College Dublin, College Green, Dublin, Ireland, 5 Centre for the Study of Existential Risk, University of Cambridge, Cambridge, United Kingdom

उ These authors contributed equally to this work.
* il291@cantab.ac.uk

## Abstract

Large host–virus association databases are increasingly used to explore broad questions in disease ecology, particularly around host range, pathogen diversity, and the potential for spillover. While these databases have been instrumental in large-scale synthesis of host–pathogen biogeography and zoonotic risk, their potential role in addressing fine-scale questions about pathogen prevalence, maintenance, and transmission dynamics remains underexplored. In this study, we build on previous efforts to assess how different types of data, including both entries in databases and the original studies they draw from, can support targeted research on zoonotic spillover. We selected two zoonotic diseases, Ebola virus disease and Lassa fever, which are characterised by recurrent spillover events and outbreaks in sub-Saharan Africa. We searched the VIRION database for entries corresponding to the respective viral taxa, the genus *Orthoebolavirus* and the species *Mammarenavirus lassaense*, and used these entries as case studies. We evaluated the extent to which databases capture crucial contextual metadata, such as spatial and temporal resolution, negative results, and measures of viral load. Guided by a conceptual framework of factors that lead to spillover, we demonstrate that while host–virus databases are valuable for addressing high-level patterns, fine-scale investigations of spillover require specific studies with detailed epidemiological data. Our study adds to a growing body of literature offering practical recommendations for database users and managers and highlights how these tools can be used as starting points in spillover research.

## Author summary

Viruses and their host species are documented in large, existing databases of host–pathogen associations. We wanted to understand how these databases

**Data availability statement:** The pipeline for creating VIRION, the original source, can be found here: https://github.com/viralemergence/virion. The derived filtered dataset can be found in S1 Appendix.

**Funding:** MMJ was supported by a Gates Cambridge Scholarship enabled by grant OPP1144 from the Bill & Melinda Gates Foundation. The funders had no role in study design, data collection and analysis, decision to publish, or preparation of the manuscript.

**Competing interests:** The authors have declared that no competing interests exist.

can help us study the risk of diseases spreading from animals to humans, a process known as spillover. These databases are often used to answer big-picture questions, such as which animals carry the most viruses or where new diseases might emerge. However, it is not yet clear how useful these databases are for answering more detailed questions, such as how often a virus infects a particular species, or how it spreads between animals and humans.

In this study, we scrutinised the information relevant to spillover in large databases and in the original research studies included in those databases. Focusing on the examples of Ebola and Lassa diseases in Africa, we found that databases are effective for spotting large patterns, but they often lack the detail needed for more focused investigations. To study spillover in depth, we need more specific information from individual scientific studies.

Our work offers practical suggestions for researchers and database managers on how to make better use of these tools and how they can serve as useful starting points for more detailed disease research.

## Introduction

Large public databases provide comprehensive evidence to answer classic and contemporary research questions about complex environmental and epidemiological phenomena [1,2], serving as a powerful complement to traditional systematic reviews and meta-analyses [3]. The last 20 years have seen growing concern about emerging infectious diseases, particularly those of zoonotic origin, and the potential for climate change and other anthropogenic factors to increase the risk of outbreaks. To understand recent global trends in zoonotic pathogen dynamics, several initiatives have curated and utilised host–virus association databases, collating published records of viral detections in human or animal hosts with detailed taxonomic information, along with selected metadata (e.g., detection methods, collection date, and location). The Global Virome in One Network (VIRION) database is currently the largest open-access host–virus association database [4]. VIRION compiles known host–virus associations, which are drawn from harmonised and reconciled aggregations of individual studies from large static databases (GMPD2, HP3, Shaw, and EID2, which together form the CLOVER database) and combined with data extracted daily from a dynamic source (GenBank). Other databases include the recently-launched PHAROS initiative (pharos.viralemergence.org, March 2022), which includes real-time updates of new pathogen detections, reducing the time lag associated with studies being published and aggregated. One of the biggest benefits of PHAROS is the inclusion of negative data [5,6], which is not included in VIRION. The ZOVER database [7] is also used in disease ecology research; although its taxonomic scope is limited to bat and rodent viruses, it remains a valuable resource for addressing questions at fine spatial and temporal resolutions. Combined with cutting-edge statistical and modelling

methods, these databases facilitate broad-scale research on zoonotic risk, for example, predictive studies of host status [8,9]. In contrast, there is more uncertainty about the potential contribution of these databases to research on spillover and its underlying mechanisms.

Plowright et al. provide a holistic framework for zoonotic spillover research [10], splitting the spillover pathway into a "hierarchical series of barriers", factors that inhibit the flow of a pathogen from reservoir host to recipient host (e.g., host misalignment in space and time, environmental factors that limit pathogen survival). Spillover can only occur when barrier gaps align spatially and temporally to create a transmission pathway. Identifying every barrier and potential preventative intervention for a pathogen requires integrating large amounts of information, often from multiple disciplines (e.g., ecology, virology, and clinical medicine), which can rarely be coordinated into a single research initiative. Guided by this framework, we explore the feasibility of using host–virus association databases to consolidate some of this information and address questions about different stages of the transmission pathway.

For this study, we chose two emerging zoonotic diseases of major concern in Africa as case studies: viral haemorrhagic fever caused by viruses in the *Orthoebolavirus* genus (family *Filoviridae*) [11], and Lassa fever, caused by *Mammarenavirus lassaense* (family *Arenaviridae*) [12], because they exemplify the complex challenges associated with understanding viral spillover. In the case of orthoebolaviruses, there remain significant unanswered questions regarding the reservoir host(s) and transmission mechanisms of the six currently known species in this genus [13–15]. These uncertainties make this genus an ideal candidate for exploring how existing data can, or cannot, be used to address critical gaps in our understanding of spillover dynamics. *Mammarenavirus lassaense*, while somewhat better characterised, still remains understudied in terms of host ecology and transmission patterns. The primary reservoir is thought to be wild rodent *Mastomys natalensis*; however, the contributions of other rodent taxa to *M. lassaense* maintenance remain unclear [16]. Importantly, these viruses are the focus of sustained surveillance efforts [17], particularly in regions where they are known to be endemic. This existing surveillance infrastructure suggests that sufficient data may be available to enable robust analysis. Focusing on these two viral taxa, we interrogated the opportunities and limitations of host–virus association databases in answering high priority questions about zoonotic emergence.

Our study is structured around three questions at narrowing scales of inquiry: *Which hosts are involved in spillover? What is the pathogen prevalence across space and time? How is the pathogen maintained and shed by the host?* These questions have been structurally adapted from the framework proposed by Plowright et al. [10], which outlines the key stages of the spillover pathway upstream of human exposure to a known pathogen. For each question, we considered the type of data necessary to produce meaningful answers, and we evaluated whether these data can be sourced from existing host–virus association databases or whether additional targeted, original research is required. While it is evident that these questions go beyond the simple fact of knowing which viruses infect which hosts (i.e., the stated purpose of host–virus association databases), our goal was to assess whether other variables and metadata contained in these databases held sufficient evidence to investigate these questions and, if not, what barriers might prevent relevant information from being recorded. We chose to focus on VIRION as it is the largest, most comprehensive open-access database of its kind. It compiles and harmonises relevant datasets and reconciles taxonomy to a single consistent backbone, so the data are accessible and standardised. However, our goal is to draw lessons applicable to any host–pathogen database and maximise their potential to support future research on spillover.

## Methods

To avoid misinterpretation in this study, we employ a commonly accepted definition of *zoonotic pathogen*, which is "a pathogen that is harboured and transmitted from vertebrate nonhuman animals to humans" [10]. A general definition of *spillover*, used here, is the "cross-species transmission of a pathogen into a host population not previously infected" [18]. We define a reservoir source population to be "one or more epidemiologically connected populations or environments in which a pathogen can be permanently maintained and from which infection is transmitted to the target population" [19,20].

When using VIRION or the original studies cited within VIRION to investigate *Orthoebolavirus* and *M. lassaense* detections, we filtered the database (accessed February 21, 2025) for NCBI-resolved hosts and separately for any NCBI-resolved *Orthoebolavirus* (*n* = 1,764 detections) and *M. lassaense* (*n* = 876 detections) species. For *Orthoebolavirus* detections, once we produced a final filtered dataset, we then read 40 of the 42 original studies in the column "ReferenceText" and extracted metadata from each study (two studies could not be accessed). The details of all 40 papers, including dates and locations of sampling, number of individuals sampled, number of pathogens sampled for, number of species sampled, type of study, detection method, and positive and negative results are summarised in the results section and fully listed in S1 Appendix. Out of these 40 papers, 19 were surveillance studies, 13 were experimental studies, six were scientific review papers, and two were modelling papers. Only the 19 surveillance studies constitute new evidence of viral detection. Of the total 40 *Orthoebolavirus* studies included in VIRION, only one entry inaccurately reports the pathogen detected in the original study. This GMPD2 entry incorrectly labelled the detection of *Orthoebolavirus taiense* in a chimpanzee [21] as *Orthoebolavirus zairense*.

## Results

### Dataset description

There were 1,764 entries for *Orthoebolavirus* and 876 entries for *M. lassaense* in VIRION. For both viral clades, entries include resolved host–virus associations, detection methods, and information on the source database. However, collection date information is missing for 140 of the 1,764 *Orthoebolavirus* entries and 185 of the 876 *M. lassaense* entries. Temporal data correspond exclusively to GenBank-derived records, as the other databases within VIRION do not report collection date or year.

None of the entries in either viral clade report negative results, geographical source of detection, or sample type in VIRION, as these variables are outside the scope of the contributing databases. The majority of records include an NCBI accession number (94.6% of *Orthoebolavirus* entries and 90.6% of *M. lassaense* entries). In contrast, relatively few entries reference the original study (7.7% of *Orthoebolavirus* entries and 20.4% of *M. lassaense* entries). These entries are from the GMPD2, HP3, Shaw, and EID2 databases that feed into VIRION.

From the 42 *Orthoebolavirus* studies identified in VIRION, 19 presented new viral detection data and were therefore examined in greater detail. These studies were assessed to determine whether they reported additional metadata not collected by VIRION, including collection date, geographic resolution, study design complexity, and detection methodology. While some studies reported detailed sample-level information, reporting was inconsistent across metadata categories, highlighting both the value and the limitations of relying on primary literature to supplement database-derived records (Table 1).

Having examined the completeness of metadata in both VIRION and the original studies referenced within it, we now outline the potential ways in which a host–virus association database could be used to answer three questions across progressively narrower scales of inquiry.

### Question 1 (broad scale): Which hosts are involved in spillover?

VIRION provides a high-level overview of the known host range of a virus. A first requirement for identifying a reservoir source population is detecting the virus within the host. Host–pathogen association databases list all the hosts that a pathogen has been detected in by taxonomic group, an especially useful feature given that review papers only sometimes provide full lists of known host species. The VIRION database also harmonises host species to NCBI taxonomy, allowing for standardised reporting. In the final filtered VIRION dataset, there were 31 host species associated with *Orthoebolavirus* from 12 families and five mammalian orders (Artiodactyla, Chiroptera, Eulipotyphla, Rodentia, Primates) [4].

**Table 1. Availability of additional metadata in *Orthoebolavirus* studies (*n* = 19) referenced in VIRION.**

| Data or Metadata | Number of studies reporting (%) |
|---|---|
| **New *Orthoebolavirus* detections** | **19 (100)** |
| **Temporal data (any)** | **18 (94.7)** |
| − **Collection date** | **5 (26.3)** |
| − **Collection year or date range** | **13 (68.4)** |
| **Location reported (country level)** | **19 (100)** |
| − Coordinates | 5 (26.3) |
| − Locality | 7 (36.8) |
| − State | 7 (36.8) |
| **Multiple viruses tested in study** | **13 (68.4)** |
| **Multiple host species tested in study** | **7 (36.8)** |
| **Multiple detection methods used** | **12 (63.2)** |
| **Negative data included for each detection method, host and virus** | **15 (78.9)** |

A host–virus association database is a powerful tool for identifying potential co-infections, which may influence infection intensity and dynamics within hosts. Co-infection with multiple pathogens can alter immune responses, shedding patterns, and transmission potential [22,23]. Six of 19 *Orthoebolavirus* studies referenced in VIRION also tested for other viruses (S1 Appendix). These studies highlight patterns of viral co-circulation and guide further targeted surveillance.

As a comparison, there are 22 host species associated with *M. lassaense* in the filtered VIRION dataset. Some of the host taxa included in VIRION reflect laboratory models including the guinea pig (*Cavia porcellus*) and the crab-eating macaque (*Macaca fascicularis*). Several other host taxa associated with *M. lassaense* in VIRION likely represent serological cross-reactivity from related mammarenaviruses outside the geographic distribution of *M. lassaense* including Asiatic bandicoot rats (genus *Bandicota*). Databases like ZOVER incorporate relevant metadata, thanks in part to the efforts of researchers. Simons et al. completed a systematic review and synthesised data from all published surveillance for *M. lassaense* in African rodents [12]. This revealed credible reports of viral associations with 12 rodent species and two additional genera (Fig 1).

### Question 2 (medium scale): What is the pathogen prevalence across space and time?

Plowright et al. highlight how infection prevalence is key to determining the distribution and intensity of infection in time and space [10]. To answer medium-scale questions, such as understanding *Orthoebolavirus* prevalence across space and time, researchers require more detailed data than what is reported in VIRION. Quantifying prevalence requires both positive and negative test results, as well as robust spatial and temporal metadata.

Negative results provide necessary context for interpreting where and when a pathogen has *not yet* been detected. Host–virus association databases theoretically allow for comparisons of the number of individuals tested for a particular virus and the total number that test positive. This positivity rate provides an estimate of pathogen prevalence within a population. However, this is only possible if negative results are reported alongside positive results. The Han et al. [8] and Schmidt et al. [24] *Orthoebolavirus*-specific datasets include this information, though presence-only databases (e.g., VIRION) do not. For each of the 19 *Orthoebolavirus* surveillance studies in VIRION, we manually extracted the number of samples collected and the subsequent proportions of positive, negative, or inconclusive results (Fig 2). Obtaining these data required consulting the original

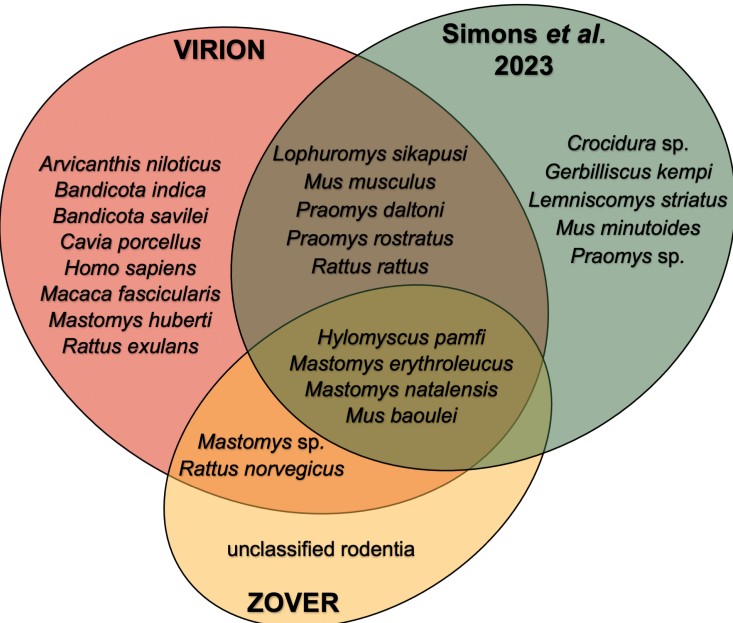

**Fig 1.** *Mammarenavirus lassaense* hosts as reported in VIRION, ZOVER, and Simons et al. [12].

studies, where the information was often reported in supplementary materials and consequently more difficult to locate and access.

To estimate variations in pathogen prevalence across space and time, GenBank stands out as the most reliable resource among the databases integrated into VIRION. While VIRION includes collection date for GenBank entries, spatial and temporal metadata are inconsistently reported for other contributing databases such as EID2, HP3, Shaw, and GMPD2. Our manual review of 19 *Orthoebolavirus* surveillance studies in VIRION revealed missing or incomplete spatiotemporal information in the database, limiting the usefulness of VIRION for tracking consistent viral detections across host species, time, and locations (S1 Appendix). For example, Pourrut et al. report repeated *O. zairense* antibody detections in *Myonycteris torquata* across multiple years and locations [25], which were not each separately captured in VIRION. GenBank, in contrast, consistently reports country-level data and includes specific geolocation for approximately 20% of entries, with GMPD2 reporting coordinates for ~98% of its records. GenBank also includes collection dates for the majority of entries, though the level of precision varies from specific dates to year only. This is particularly important when investigating seasonal trends, such as fluctuations in bat shedding associated with reproductive cycles [26–28]. However, a clear limitation of GenBank is its lack of serological data, which can help infer prior exposure in potential host taxa in lieu of detecting pathogen genetic material; this is especially relevant for viruses with low-amplitude epidemic curves.

Despite a paucity of pre-2014 data, GenBank currently contains over 4,000 *Orthoebolavirus* sequences (92% of entries in VIRION are from GenBank), including 332 non-human entries, offering a stronger foundation for spatial and temporal analysis than most aggregated databases. While fine-scale geographic data are ideal, especially for fragmented host populations where viral presence may vary across ranges [29,30], even coarse data from GenBank may improve our understanding of exposure risk and transmission dynamics.

For *M. lassaense*, in contrast, more progress has been made toward making such data accessible. 695/876 (79.3%) of the *M. lassaense* entries are from Genbank. However, critically, work by Simons et al. resulted in an aggregate database with fine scale spatial and temporal resolution as well as inclusion of negative testing data, allowing for a true evaluation

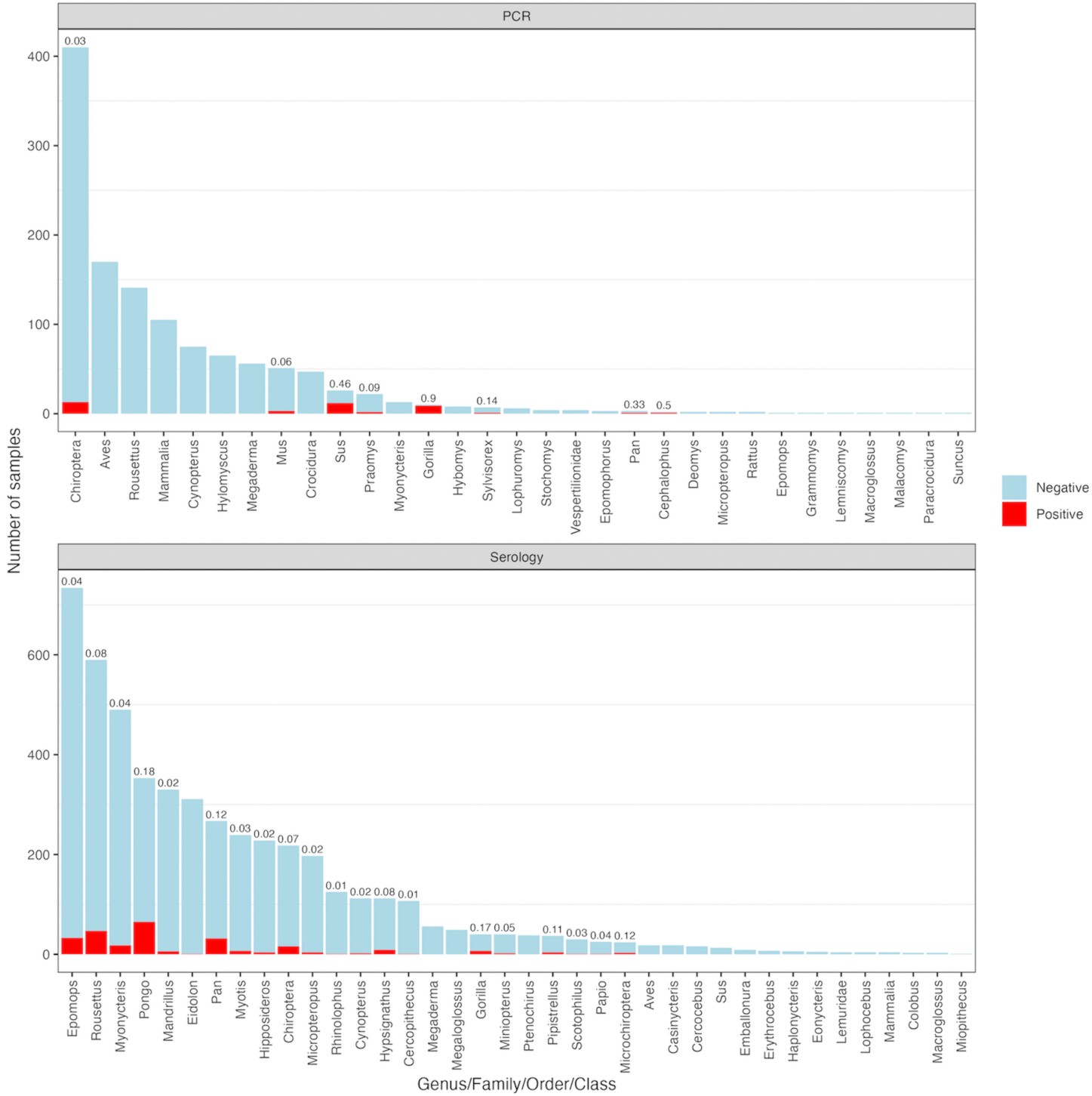

**Fig 2. Positive and negative non-human samples reported in the 21 original *Orthoebolavirus* surveillance studies referenced in the filtered VIRION dataset.** Numbers on top of bars indicate positivity rates for taxonomic groups with at least one positive sample. Samples were categorized by genus or the next finest taxonomic resolution available. Viral isolation samples were not reported, as isolation is typically only attempted on PCR-positive samples. Results that were not positive were noted as negative or inconclusive following Han et al. [8], due to many studies not specifying result details.

of sampling effort across host taxa [12]. These data synthesis efforts enable the investigation of finer-scale questions across space and time.

### Question 3 (fine scale): How is the pathogen maintained and shed by the host?

Plowright et al. suggest that pathogen release is an important bottleneck in the spillover pathway, whereby it is a barrier that permits or constrains the flow of pathogens from one species to another [10]. They highlight how the release of a pathogen is linked to its "presence and viability in relevant tissues" which has downstream impacts on human exposure. Questions about the shedding and maintenance of a pathogen by individual host species require fine-grained data and can only be addressed through detailed epidemiological or experimental studies designed specifically to capture these dynamics. This careful and thorough investigation is particularly critical in unravelling the sources of unknown disease outbreaks, even when a certain zoonotic origin is suspected or presumed from the outset [31].

Data on viral load and tissue tropism are critical for answering these questions, as they provide clues about infectiousness, routes of transmission, and potential reservoir status. The risk of human-to-human *O. zairense* transmission increases with viral load, which is greatest in the later stages of illness and immediately after death [32]. Viral load is defined as the quantity of virus in a volume of fluid at a given time and is quantified by the number of PCR cycles (Ct value) required to achieve detection [33]. For meaningful comparisons, RNA extraction and qRT-PCR procedures must be run with a standard curve [34]. However, Ct values only measure the amount of genetic material for the qPCR target and may reflect gene expression or genome copies, therefore not directly correlating to the amount of infectious virus present. Direct measurements of infectious virus, such as plaque assays or $TCID_{50}$, are more informative for assessing transmission potential [35]. Unfortunately, information on viral load is rarely available in standard surveillance datasets or even in many original pathogen detection studies. Most of the 19 *Orthoebolavirus* surveillance studies referenced in VIRION lack the resolution needed to assess shedding dynamics, as PCR detections are typically not accompanied by quantitative viral load data.

Contextual information on sample type was not found in any of the host–virus association databases we assessed. However, we obtained the sample sources from the 19 *Orthoebolavirus* surveillance studies, excluding human entries, referenced in VIRION. These studies included 10,219 detection attempts, and sample sources were reported for all of them (Fig 3). Reporting the sources of negative as well as positive samples is also useful in understanding the rate and mechanism of viral shedding within species (e.g., differences across age, sex, and seasons).

However, for certain well-studied viruses such as influenza, comprehensive epidemiological databases do exist and include fine-scale clinical and virological data, making it possible to assess maintenance and shedding patterns. This includes the World Animal Health Information System [36], WHISPers [37], and government outbreak reports, such as those published by the UK Animal and Plant Health Agency. However, for most zoonotic pathogens, particularly those that are rare or emerging, such fine-grained data remain limited. For example, entries in the VIRION database, such as for *Orthoebolavirus* and *M. lassaense,* do not report sample type.

## Discussion

### Considerations for database users

When using host–virus association databases for understanding spillover, it is crucial to clearly define the research question and consider the scale at which that question is posed. The scale, whether broad or narrow, will influence the type of data sources that are most appropriate. For broad-scale questions, particularly those concerning well-characterised and extensively studied pathogens such as influenza, host–virus databases can be immensely useful. These resources compile a vast amount of information from multiple studies, making it easier to identify general patterns, trends, and associations across different host species, time periods, or geographic regions. Question 1 illustrates and builds on two key

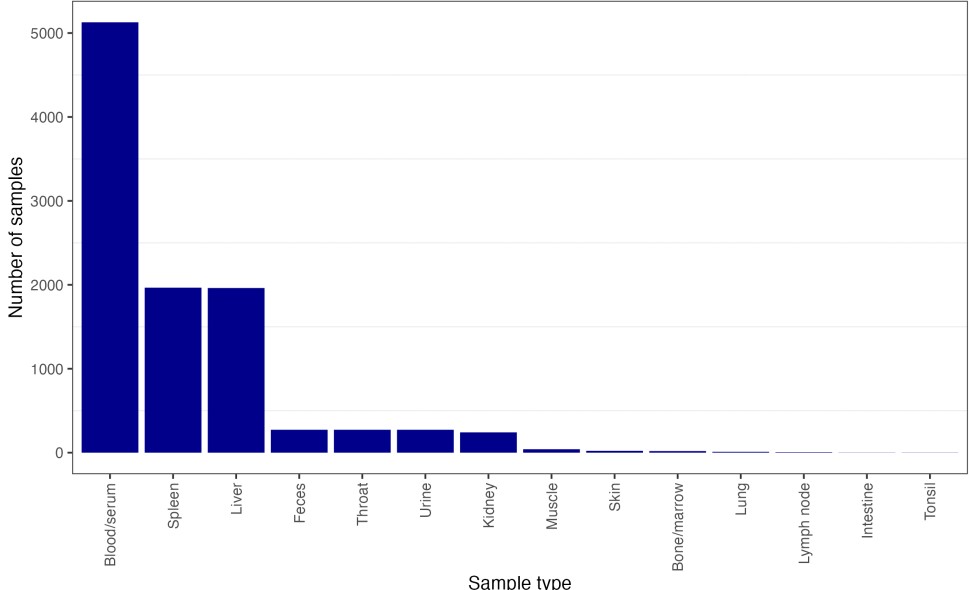

**Fig 3. Sample types of all *Orthoebolavirus* entries in VIRION, extracted from the 19 original studies and including all detection methods.**

points: firstly, understanding the system of interest is critical for interpretation of aggregated host–virus association data; secondly, researchers should carefully consider whether their approach is to infer all possible hosts (e.g., species susceptible to infection) or the most likely species to serve as reservoirs in the wild.

Broadly, host–virus associations are useful for generating initial hypotheses about which taxa may play a role in the spillover process. While they do not provide the level of detail required to establish transmission dynamics or reservoir status, they offer a broad perspective on the potential diversity of hosts and narrow down the list of species for further investigation. In the context of *Orthoebolavirus*, this allows researchers to map out candidate hosts that could be involved at various points in the spillover chain, from reservoirs to intermediate hosts, thereby guiding more targeted surveillance efforts in the field and in museum collections.

In contrast, when investigating more specific or fine-scale questions, such as those focused on particular host populations, local transmission dynamics, or under-studied pathogens, researchers must go beyond databases and analyse original data from the primary literature. In such cases, large databases still play an important role by pointing researchers towards relevant studies, but they should not be relied upon as the ultimate source of information. System-specific and metadata-rich datasets are better choices for more targeted investigations [38].

This study builds on previous work by demonstrating that it is essential to approach all data, whether from databases or original studies, with a critical mindset. Both sources are affected by biases and gaps, such as uneven geographic and taxonomic sampling, varying methodology, or publication bias. Researchers have begun addressing these biases in studies using both data in VIRION [39,40] and data not in VIRION [40]; however, many biases are not currently accounted for in host–virus association databases. Below, we describe three important biases and illustrate them with findings from our study.

1. **Host taxonomic bias and uncertainty:** Focusing on a single species or taxonomic group may perpetuate a biased belief that it plays a prominent role in spillover (e.g., as a reservoir or intermediate host) despite insufficient evidence [41]. For example, there is a well-documented "bat bias" in efforts to identify the reservoir of *Orthoebolavirus* [42]. Although conclusive evidence remains elusive [15,43], fruit bats are disproportionately represented in databases.

This is partly due to the lack of negative results in presence-only datasets like VIRION, making it difficult to determine whether observed patterns reflect true distribution or sampling effort. The first direct evidence pointing to bats as reservoir hosts of *Orthoebolavirus* was published by Leroy et al. [44], when viral RNA and antibodies were detected in three species of Old World fruit bats (family Pteropodidae). This drove an enormous sampling bias towards fruit bats [13,45]. Han et al. [8] estimated that five times as many fruit bat individuals have been sampled for filoviruses than insectivorous bats from multiple families, even though Old World fruit bats comprise only ~16% of global bat biodiversity (186/1116 extant species) [8] and just 12% of the 266 bat species in sub-Saharan Africa [46]. If this "bat bias" is unaccounted for, the involvement of fruit bats in filovirus spillover might be over-interpreted [42].

2. **Sampling and methodological biases:** Non-lethal sampling methods, such as blood, urine, faeces, and saliva collection, are often preferred for ethical and practical reasons, especially in small animals. Orthoebolaviruses can cause high levels of viremia, with viral RNA loads exceeding $10^{10}$ genome copies in nonhuman primates [47] and greater than $10^6$ viral RNA copies in Angolan free-tailed bats [48]. Blood is therefore the most common sample type in *Orthoebolavirus* detection studies (37% of the original studies referenced in VIRION), while organ or tissue samples are underrepresented. Consequently, there is an abundance of serological studies as well as RNA detection attempts (as orthoebolaviruses are able to infect cells circulating in blood, unlike other RNA viruses). Within the serological detection attempts, different serological assays have varying levels of specificity—due to cross-reactivity—and can test for different types of antibodies (e.g., IgG vs. IgM antibodies), hindering direct comparisons between studies. These challenges are further compounded by the lack of species-specific secondary antibodies for bats which limits assay sensitivity and specificity and often requires the use of heterologous reagents with unknown cross-reactivity (e.g., anti-mouse or proteins A and G). Additionally, bats infected with filoviruses do not appear to generate neutralising antibodies [49] which precludes the use of virus neutralization assays as the "gold standard" in serological surveillance for orthoebolaviruses in bats. The majority of the 141 host–virus associations within the 16 serological studies were detected by enzyme-linked immunosorbent assay (ELISA, $n = 133$) and the remainder were detected by indirect fluorescent antibody test (IFA, $n = 8$) (see S1 Appendix). Viral isolation, which requires a cold chain and maximum containment biosafety facilities, is also rarely attempted, leading to further skew in detection methods. Molecular surveillance data (e.g., PCR and viral isolation) can be interpreted as stronger evidence of active viral circulation relative to serological data. PCR identifies the presence of viral genetic material, providing robust evidence of a current or very recent infection. In contrast, serological studies detect host antibodies generated in response to prior exposure, which may persist long after the virus has been cleared. Even when investigating past infections, serological data should be interpreted with caution due to the possibility of false negative or false positive results, due to low sensitivity and low specificity, respectively.

3. **Geographical bias:** Surveillance studies are often conducted in response to outbreaks, resulting in an overrepresentation of data from affected areas where pathogen prevalence may be unusually high. Of the 19 *Orthoebolavirus* studies in VIRION, 14 were cross-sectional animal studies, and eight of these were carried out in the same location as an Ebola outbreak [25,44,50–55].

These issues significantly affect the reliability and representativeness of the data being used, potentially leading to skewed interpretations if not properly accounted for. Therefore, researchers must be cautious and transparent about these limitations in their analyses and conclusions. Thoughtful and informed use of databases, combined with careful engagement with primary literature, is key to conducting rigorous and meaningful research on host–pathogen interactions.

### Higher-level considerations for database curators, researchers, and funders

A recurring, fundamental issue is that for many pathogens, particularly those that are rare or emerging, the data required to answer key questions about spillover simply do not exist [56]. In some cases, relevant information may have been

collected, but is paywalled, poorly indexed, or not digitised at all. This presents a serious barrier to progress, particularly when studying high-consequence pathogens such as orthoebolaviruses, for which data are sparse and fragmented.

If we are to improve our capacity to understand and predict zoonotic spillover, two critical priorities must be addressed. First, there must be greater investment in data acquisition. This includes expanding and sustaining disease surveillance efforts in both human and animal populations, particularly in regions that are underrepresented in current datasets. Enhanced surveillance will help fill major data gaps and provide the foundational information needed to support robust analyses. One contributing factor to the lack of published negative results and surveillance studies in non-outbreak regions is there being a lower likelihood of obtaining positive results. The increasing pressure to publish may incentivise researchers to sample in already well-represented regions rather than in understudied areas, as negative results are difficult to publish in scientific journals [57].

Second, there is a pressing need for improved data sharing, aggregation, and curation [58]. Notably, these actions demonstrate the importance of depositing data in common databases at publication to enhance accessibility and repeatability of publicly funded research (FAIR principles) [59]. Detailed research relies on the publication of negative results as well as metadata on sample type, collection site, collection date, co-infection status, detection method, and viral load. This depends not only on infrastructure and funding, but also on a cultural shift within the research community. Researchers should be encouraged and incentivised to submit their data to shared platforms, particularly dynamic databases that allow for real-time updates and accept both positive and negative results. If a species is reported to harbour a virus across space and time, it may indicate that the host is a persistent source of infection [20]. Platforms like PHAROS are working to include negative results, and genetic sequence repositories like GenBank include temporal and geographic data that contextualise prevalence patterns. These resources enable more comprehensive analysis of pathogen prevalence across both time and space, particularly in systems where surveillance and data sharing are more advanced. Databases such as PHAROS have significant potential but remain underutilised. By contributing data in a timely and transparent manner, researchers can help build more complete and usable datasets that benefit the entire field.

Researchers can also enhance analysis, particularly of spatial and temporal patterns of spillover, by using complementary biodiversity datasets such as the IUCN Red List, GBIF, and the Living Planet Index, which provide detailed host distribution data for modelling pathogen spillover risk. While these datasets are often limited by a lack of fine-scale species occurrence data (e.g., IUCN species ranges often overstate true distributions), they offer a valuable starting point for locating complementary datasets.

Addressing these challenges will require coordinated efforts from across the scientific ecosystem, including researchers, journal editors, database managers, and funding bodies. Without meaningful improvements in both data availability and accessibility, our capacity to answer even the most basic questions about pathogen spillover will remain limited, particularly for the rare but potentially devastating viruses that pose the greatest threats to global health. In summary, our practical recommendations for researchers and database managers include: (1) systematic reporting of negative results alongside positive findings in both primary studies and databases; (2) increased sampling across underrepresented temporal and spatial scales, as well as across a broader range of species; (3) detailed reporting of sample sources in both primary studies and databases; and (4) active investigation and documentation of co-infection. Additionally, database curators should clearly define and communicate the variables captured in the database, ensuring alignment with the intended purpose of these resources and facilitating appropriate use by the research community.

Our study is limited by its reliance on a single composite dataset, derived from multiple host–virus association databases. Our exploration of the original literature was limited to *Orthoebolavirus*, resulting in a relatively small sample size, which we acknowledge as a limitation of this study. Nevertheless, similar challenges are likely to arise for other viruses and datasets, and we encourage further research to assess the generality of these findings.

## Conclusion

In summary, host–virus association databases such as VIRION, and the datasets it draws from (e.g., GenBank), can provide valuable insights into broad-scale questions about viral spillover, particularly those concerning general host range or high-level patterns of host associations. These databases are powerful tools for large-scale synthesis, enabling systematic reviews or meta-analyses. However, we documented limitations in their current suitability as primary sources of data for fine-scale questions about spillover dynamics, such as pathogen maintenance, shedding patterns, or spatial and temporal variation in prevalence, which require detailed, context-rich data. We build on previous studies and outline that major gaps—in the databases themselves as well as the original surveillance studies—include a lack of negative results, a lack of precise metadata, and methodological inconsistency. We therefore reiterate that, firstly, host–virus databases should be used primarily as entry points for locating GenBank data or identifying key surveillance studies, rather than as final, definitive sources of data for spillover research. Secondly, database curators should seek to extract and incorporate additional contextual metadata from original studies. Thirdly, researchers who collect and publish primary data on viral detection should share complete methods and results to ensure maximum value for the wider scientific and medical community. Finally, users must consider the inherent limitations and biases within these datasets. When applied appropriately, these databases can streamline the early stages of zoonotic spillover research, but finer-scale understanding necessitates targeted surveillance studies.

## Supporting information

**S1 Appendix. Summary of the metadata in the scientific papers referenced for *Orthoebolavirus* entries in the VIRION database.**
(XLSX)

## Author contributions

**Conceptualization:** Imogen C. Lindsley, Freya Jephcott, Olivier Restif.

**Data curation:** Imogen C. Lindsley, Rory Gibb, Gregory F. Albery.

**Formal analysis:** Imogen C. Lindsley.

**Investigation:** Imogen C. Lindsley, Olivier Restif.

**Methodology:** Imogen C. Lindsley.

**Supervision:** Maya M. Juman, Freya Jephcott, Olivier Restif.

**Writing – original draft:** Imogen C. Lindsley, Maya M. Juman.

**Writing – review & editing:** Imogen C. Lindsley, Maya M. Juman, Stephanie N. Seifert, Rory Gibb, Gregory F. Albery, Freya Jephcott, Olivier Restif.

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
