## [Decision Letter · Decision Letter 0]

24 Dec 2025

PNTD-D-25-01065

Host-virus association databases as tools for understanding viral spillover at varying scales

Dear Dr. Lindsley,

Thank you for submitting your manuscript to PLOS Neglected Tropical Diseases. After careful consideration, we feel that it has merit but does not fully meet PLOS Neglected Tropical Diseases's publication criteria as it currently stands. Therefore, we invite you to submit a revised version of the manuscript that addresses the points raised during the review process.

Please submit your revised manuscript within by Feb 22 2026 11:59PM. If you will need more time than this to complete your revisions, please reply to this message or contact the journal office at plosntds@plos.org. Please include the following items when submitting your revised manuscript:

We look forward to receiving your revised manuscript.

Kind regards,

Colleen B Jonsson, PhD

Academic Editor

Georgios Pappas

Section Editor

Shaden Kamhawi

co-Editor-in-Chief

Paul Brindley

co-Editor-in-Chief

**Journal Requirements:**

At this stage, the following Authors/Authors require contributions: Imogen C Lindsley, Maya M Juman, Stephanie N Seifert, Rory Gibb, Gregory F Albery, Freya Jephcott, and Olivier Restif. Please ensure that the full contributions of each author are acknowledged in the "Add/Edit/Remove Authors" section of our submission form.

3) Please send a completed 'Competing Interests' statement, including any COIs declared by your co-authors. If you have no competing interests to declare, please state "The authors have declared that no competing interests exist". Otherwise please declare all competing interests beginning with the statement "I have read the journal's policy and the authors of this manuscript have the following competing interests"

**Reviewers' Comments:**

Reviewer's Responses to Questions

**Key Review Criteria Required for Acceptance?**

**Methods**

-Are the objectives of the study clearly articulated with a clear testable hypothesis stated?

-Is the study design appropriate to address the stated objectives?

-Is the population clearly described and appropriate for the hypothesis being tested?

-Is the sample size sufficient to ensure adequate power to address the hypothesis being tested?

-Were correct statistical analysis used to support conclusions?

-Are there concerns about ethical or regulatory requirements being met?

Reviewer #1: The data are described clearly, and are appropriate for the study. The methods are also described clearly and appropriate for the study.

Reviewer #2: The study design and methods are clearly described, but the analytical depth is limited. The approach primarily involves filtering the VIRION database for Orthoebolavirus entries and manually extracting metadata from ~40 source studies (of which 19 are surveillance studies). While the effort to cross-check and verify metadata is useful, the scope of the dataset is too narrow to support broader inferences about database performance or spillover mechanisms.

There is no quantitative or comparative analysis included — only descriptive summaries. The so-called “experiment” mentioned in the resubmission letter refers to manual data review and does not constitute an analytical or empirical experiment. To strengthen this section, the authors could include some quantitative assessment of metadata completeness (e.g., percentage of studies reporting negative data, spatial coordinates, or detection methods). That would provide an objective measure of data quality and improve the credibility of the “fine-scale” assessment they aim to demonstrate.

**Results**

-Does the analysis presented match the analysis plan?

-Are the results clearly and completely presented?

-Are the figures (Tables, Images) of sufficient quality for clarity?

Reviewer #1: Yes, to all three questions.

Reviewer #2: The results section primarily summarizes what is already visible from the databases and the original studies. The figures (1–3) are descriptive — listing host species, reporting positive and negative samples, and showing sample types. These are useful visual aids but do not offer new insights or analyses.

Importantly, the number of studies (40 total, 19 with new detections) is too small to claim that the analysis tests database “utility at varying scales.” The conclusions drawn — that negative results and metadata are often missing, and that databases are better suited for large-scale rather than fine-scale analyses — are correct but already well established in the literature (e.g., Carlson et al. 2022; Gibb et al. 2021; Simons et al. 2023).

The section also mixes interpretation and guidance (e.g., “researchers can enhance these analyses using complementary biodiversity datasets”) into what should be reported results. Separating these into results versus recommendations would improve clarity. A quantitative summary table showing the frequency of missing fields or metadata would make the results section more substantial and analytical.

**Conclusions**

-Are the conclusions supported by the data presented?

-Are the limitations of analysis clearly described?

-Do the authors discuss how these data can be helpful to advance our understanding of the topic under study?

-Is public health relevance addressed?

Reviewer #1: The conclusions are supported by the data, methods and results presented. I ask the authors to consider my comment on the manuscript that what this study has done is to document (not identify) the limitations of "big data" for fine-scale studies.

Reviewer #2: The conclusions are reasonable and align with the descriptive findings but somewhat overstate the originality of the study. The main takeaway — that host–virus databases are valuable for broad-scale questions but limited for fine-scale spillover analyses — is valid yet not new. This has been repeatedly discussed in prior database evaluations.

The recommendations (for better metadata standards, inclusion of negative results, and consistent data sharing) are relevant and well argued but belong more to a perspective or data note than a research paper. The manuscript would therefore fit better as a Data Resource or Database Analysis Note, emphasizing transparency, metadata coverage, and lessons learned rather than as a full research article.

**Editorial and Data Presentation Modifications?**

Reviewer #1: No suggestions for this.

Reviewer #2: (No Response)

**Summary and General Comments**

Reviewer #1: My general sense of this manuscript is that the authors are demonstrating well, something that is already well understood by the research community--broad-scale databases with limited or inconsistent metadata are inadequate for answering fine-scale questions. This study documents that well, and has merit, and will probably be cited frequently. Nevertheless, the tone or the paper is that they are covering new ground and making a novel discovery, and this is really not the case.

Reviewer #2: Overall, the work is clearly written and conceptually sound but lacks analytical rigor and novelty. Expanding it with a quantitative assessment of metadata completeness, or reframing it as a data-focused paper, would strengthen its contribution and make it more appropriate for publication.

PLOS authors have the option to publish the peer review history of their article (what does this mean?). If published, this will include your full peer review and any attached files.

Reviewer #1: **Yes:**Robert D. Owen

Reviewer #2: No

**Figure resubmission:**
---

## [Decision Letter · Decision Letter 1]

8 Apr 2026

PNTD-D-25-01065R1

Host-virus association databases as tools for understanding viral spillover at varying scales

Dear Dr. Lindsley,

Thank you for submitting your manuscript to PLOS Neglected Tropical Diseases. After careful consideration, we feel that it has merit but does not fully meet PLOS Neglected Tropical Diseases's publication criteria as it currently stands. Therefore, we invite you to submit a revised version of the manuscript that addresses the points raised during the review process.

We look forward to receiving your revised manuscript.

Kind regards,

Colleen B Jonsson, PhD

Academic Editor

Georgios Pappas

Section Editor

Shaden Kamhawi

co-Editor-in-Chief

Paul Brindley

co-Editor-in-Chief

**Journal Requirements:**

**Reviewers' comments:**

Reviewer's Responses to Questions

**Key Review Criteria Required for Acceptance?**

**Methods**

-Are the objectives of the study clearly articulated with a clear testable hypothesis stated?

-Is the study design appropriate to address the stated objectives?

-Is the population clearly described and appropriate for the hypothesis being tested?

-Is the sample size sufficient to ensure adequate power to address the hypothesis being tested?

-Were correct statistical analysis used to support conclusions?

-Are there concerns about ethical or regulatory requirements being met?

Reviewer #1: As in the earlier version, Methods are appropriate, and explained clearly.

Reviewer #3: The objectives are clearly stated in this manuscript, as are the methods used to address the questions asked.

**Results**

-Does the analysis presented match the analysis plan?

-Are the results clearly and completely presented?

-Are the figures (Tables, Images) of sufficient quality for clarity?

Reviewer #1: As in the earlier version, Results seem reasonable, not unexpected.

Reviewer #3: The results are clearly presented.

**Conclusions**

-Are the conclusions supported by the data presented?

-Are the limitations of analysis clearly described?

-Do the authors discuss how these data can be helpful to advance our understanding of the topic under study?

-Is public health relevance addressed?

Reviewer #1: The conclusions are better presented than in the earlier version, and are reasonable; again, not unexpected.

Reviewer #3: The conclusions are generally supported by the data.

**Editorial and Data Presentation Modifications?**

Reviewer #1: A couple of minor issues came to mind in this reading:

1. (Lines 460, 463) I have no idea what the term "real-time" means in the context of either of these statements. Please clarify and/or use a different term.

2. (Line 472) The IUCN Red List quite often over-states a species' distribution; it should not be taken as a definitive distribution description of host species.

Reviewer #3: N/A

**Summary and General Comments**

Reviewer #1: (Lines 476-481) These are certainly good recommendations, but it seems unlikely that they will have much or any impact on the research and database communities. Can the authors offer any more practical recommendations, such that the publication of this paper will have a practical impact, rather than just confirming the known limitations of data mining?

Reviewer #3: The revised manuscript by Lindsley and Juman et al. describes the use of the VIRION database for addressing questions regarding the distribution and transmission of ebolaviruses and Lassa viruses. Overall, the manuscript is well-presented and addresses an important aspect of data harvesting. A few modifications would improve the discussion section:

1. The authors discuss shortcomings in the published/deposited data. To this discussion, they could add that there can be concerns regarding false positive and false negative serology due to cross-reactivity or low sensitivity, respectively.

2. A brief discussion of how researchers should interpret surveillance data from PCR versus serology would be useful.

PLOS authors have the option to publish the peer review history of their article (what does this mean?). If published, this will include your full peer review and any attached files.

**Do you want your identity to be public for this peer review?** For information about this choice, including consent withdrawal, please see our Privacy Policy.

Reviewer #1: **Yes:**Robert D. Owen

Reviewer #3: No

**Figure resubmission:**
---

## [Editor Report · Decision Letter 2]

4 May 2026

Dear Ms Lindsley,

We are pleased to inform you that your manuscript 'Host-virus association databases as tools for understanding viral spillover at varying scales' has been provisionally accepted for publication in PLOS Neglected Tropical Diseases.

Best regards,

Colleen B Jonsson, PhD

Academic Editor

Georgios Pappas

Section Editor

Shaden Kamhawi

co-Editor-in-Chief

Paul Brindley

co-Editor-in-Chief

---

## [Editor Report · Acceptance letter]

Dear Ms Lindsley,

We are delighted to inform you that your manuscript, "Host-virus association databases as tools for understanding viral spillover at varying scales," has been formally accepted for publication in PLOS Neglected Tropical Diseases.

Best regards,

Shaden Kamhawi

co-Editor-in-Chief

Paul Brindley

co-Editor-in-Chief
